# Pattern Analysis of Serum Galectins-1, -3, and -9 in Breast Cancer

**DOI:** 10.3390/cancers15153809

**Published:** 2023-07-27

**Authors:** Avery Funkhouser, Hayden Shuster, Julie C. Martin, W. Jeffery Edenfield, Anna V. Blenda

**Affiliations:** 1Department of Biomedical Sciences, University of South Carolina School of Medicine Greenville, Greenville, SC 29605, USA; 2Prisma Health Cancer Institute, Prisma Health, Greenville, SC 29605, USA

**Keywords:** galectin, serum, breast neoplasm, molecular subtype, galectin-1, galectin-3, galectin-9

## Abstract

**Simple Summary:**

This study aims to understand the role of galectins by breast cancer subtype and their change in response to cancer treatment. Galectins are proteins involved in cancer growth, metastasis, immune evasion, and cell division. Galectin-1, -3, and -9 levels were measured in breast cancer patients using a technique called ELISA and analyzed for their relationship with tumor characteristics such as stage, subtype, and receptor expression. The study found that galectin-9 levels were significantly increased in HER2-enriched tumors but reduced in hormone-receptor-positive tumors, while galectin-1 levels were higher in patients who underwent systemic cancer therapy. These findings provide valuable insights into galectin changes during cancer progression, treatment response, and tumor biology. They have implications for future research on therapeutic targets using galectin inhibitors and the use of galectin biomarkers for diagnosing and monitoring breast cancer.

**Abstract:**

Galectins have been shown to have roles in cancer progression via their contributions to angiogenesis, metastasis, cell division, and the evasion of immune destruction. This study analyzes galectin-1, -3, and -9 serum concentrations in breast cancer patients through enzyme-linked immunosorbent assay (ELISA) against the characteristics of the patient and the tumor such as stage, molecular subtype, and receptor expression. Galectin-9 was found to be statistically significantly increased in HER2-enriched tumors and reduced in patients with hormone-receptor-positive tumors. Galectin-1 was found to be statistically significantly increased in the serum of patients who had undergone hormonal, immunotherapy, or chemotherapy. These findings provide insight into the changes in galectin levels during the progress of cancer, the response to treatment, and the molecular phenotype. These findings are valuable in the further understanding of the relationships between galectin and tumor biology and can inform future research on therapeutic targets for galectin inhibitors and the utility of galectin biomarkers.

## 1. Introduction

Breast cancer is a significant public health issue in the United States, with an estimated 339,250 new cases in 2022 alone. While the mortality rates of breast cancer have consistently declined in the 21st century, the disease still leads to tens of thousands of deaths annually, with an estimated 43,250 in 2022 [1]. The impact of breast cancer on individuals and society is substantial, affecting not only the physical health and wellbeing of those diagnosed but also their families, communities, and the healthcare system [2,3]. As such, understanding the risk factors, screening and diagnostic tools, and treatment options for breast cancer is critical to reducing its impact on public health.

Breast cancer can be classified into four subtypes—Luminal A, Luminal B, HER2-enriched, and Triple Negative—based on Estrogen Receptor (ER), Progesterone Receptor (PR), and HER2 (also known as ERBB2) expression [4]. These subtypes have clinical implications regarding the type of treatment selected and the overall outcomes [5,6,7].

Galectins (Gals) are a subfamily of lectin proteins with a conserved carbohydrate-recognition domain (CRD) capable of binding ß-galactoside glycoconjugates [8]. There are three subgroups of galectins classified by their structure. Galectin-1 is in the prototypical group that also includes galectin-2, -5, -7, and -10, which are able to form dimers [9]. The sole member of the second subgroup is the chimeric galectin-3, which is widely expressed in multiple organ systems, including immune, epithelial, and neuronal tissues [10]. The final subgroup is the tandem-repeat galectins, with two CRDs that are more potent than previous groups regarding biologic response and include galectin-4, -8, -9, and -12 [11]. These proteins bind to specific carbohydrate structures on the surface of cells, thereby regulating cell adhesion, migration, signaling, and death [12,13,14,15].

Galectins are expressed in a wide variety of tissues and cell types, and their functions are highly context-dependent [16]. These biochemical functions have made galectins of interest in oncology. Galectin-1, -3, and -9 are the most extensively studied and are implicated in alterations in oncogenic pathways, apoptosis, T-cell immune response, and metastasis, dependent upon specific galectin and glycoprotein interactions [17]. Galectin-1 and -3 are both capable of binding to CD7 and CD45 respectively, and inducing apoptosis [18]. Elevated serum galectin-1 levels have been shown to decrease TH1 responses in nonmalignant cells, favoring TH2 cytokine profiles which promote tumor survival [19]. Cancer metastasis is promoted by elevated galectin-3 levels interacting with MUC1, a transmembrane protein on most secretory epithelia, limiting its protective effects due to increased interactions with CD44 and the ligands for e-selectins [20]. Galectin-3 has also been shown to have oncogenic properties through its interaction with RAS, the RAS pathway, and BCL-2 [21]. Galectin-9 has also demonstrated the ability to aid in the avoidance of cancer cell recognition through mediating TH-1 helper-cell apoptosis [22]. Therefore, there are multiple mechanisms by which galectins play a role in oncogenesis.

With these mechanisms in mind, there have been multiple studies examining galectin expression levels in various cancer tissues and the sera of cancer patients using immunohistochemistry and enzyme-linked immunosorbent assays (ELISAs) and recognizing their impact on oncological behavior [23,24,25,26,27,28,29,30,31]. The concentrations of galectins-1, -3, and -9 have been shown to be increased in the serum of breast cancer patients compared to healthy controls [32]. Targeting galectins has been demonstrated to have an impact on cancer progression [25]. Trials have examined these mechanisms to identify potential therapeutic targets to treat multiple cancer types [21,33,34,35].

While a previous study has shown that different galectins’ mRNA expression can be increased or decreased in breast cancer and that their expression in the tumor tissue changes based on the breast cancer’s molecular subtype [36], no study has shown how the corresponding circulating level of galectins changes based on molecular subtype. This study reports the serum concentrations of galectin-1, -3, and -9 in breast cancer patients organized by molecular subtype, as well as a broad screening of other data points such as hormone receptor (HR) expression levels and exposure to treatment. This provides a better characterization of the changes in galectins in breast cancer by delivering a more nuanced understanding of the factors that shape their expression in the serum. This study joins efforts of others seeking biomarkers of diagnostic and prognostic potential in a family of proteins actively involved in oncological processes.

## 2. Materials and Methods

A heterogeneous and random selection of a total of 139 serum samples from breast cancer patients was obtained from the Prisma Health Cancer Institute (PHCI) biorepository (Greenville, SC, USA). The sample collection years ranged from 2012 to 2022. Information about the nature of the PHCI and its standard operating procedures has been previously described [37].

Patient information was collected from the PHCI database. The information included sex; ethnicity; race; principal diagnosis; TNM staging; overall staging; histology; primary versus metastatic sample; tissue site of resection; specimen age; specimen collection year; patient age at collection; patient smoking history; ER, PR, and HER2 IHC assays; ER and PR percent expression on IHC staining; and systemic therapy. Systemic therapy was defined as being exposed to any hormonal therapy, chemotherapy, or immunotherapy prior to serum sample collection. Therefore, patients may have received no, one, or multiple therapies from within these therapy types. Information about treatment after sample collection was not available for this study. In rare cases, some patients did not have information documented, such as race or TNM staging. The samples lacking information were excluded from the analysis. Samples that expressed either estrogen receptor, progesterone receptor, or both were additionally categorized as hormone receptor (HR) positive.

The subtypes of the cancers were classified using the information about the ER, PR, and HER2 IHC expression data using the following method: Luminal A (strong ER+, HER2±), Luminal B (weak to moderate ER+, HER2±), HER2-enriched (ER−, PR−, HER2+), and Triple Negative (ER−, PR−, HER2−). ER ICH percent expression greater than 66% was considered “strong”. There was an absence of Ki-67 data, and classification was based on methods previously described [38].

Seventy-nine (57% of total) samples were classified as Luminal A (median age 59, min-max 19–89), eighteen (13% of total) samples were classified as Luminal B (median age 52, min-max 34–74), ten (7% of total) samples were classified as HER2-enriched (median age 60, min-max 44–77), and thirty-two (23% of total) samples were classified as Triple Negative (median age 55, min-max 30–80). The number of samples from stages I, II, III, and IV were 55, 51, 22, and 11, respectively. Further description of the number of patients and their galectin concentrations in various comparisons is available in Appendix A.

### 2.1. Galectin Profiling

The patients’ sera were used to determine circulating galectin concentrations using an enzyme-linked immunosorbent assay (ELISA) as previously described [32]. Of the 139 total samples in this study, 40 were previously used in the study by Blair et al. (2021). Concentrations of galectins-1, -3, and -9 were obtained using the ELISA kits from R&D Systems (Minneapolis, MN, USA). Of the 139 total samples, serum concentrations of galectins-1 and 3 were obtained in 138 samples, and of galectin-9 in 77 samples.

### 2.2. Data Analysis

All statistical analyses were performed using JMP^®^ 17.0 software by the SAS Institute (Cary, NC, USA). The distributions of each of the galectin concentrations were compared based on the data of interest. Not all samples had data for every variable. For example, not all patients had a smoking history recorded. Samples with missing data were excluded from analyses requiring that data. Two-group comparisons were carried out using a *t*-test. Multiple comparisons were performed with one-way ANOVA, with subsequent analysis of all pairs using Student’s *t*-test if the ANOVA’s probability > F was less than 0.05. Combination analysis of the galectin concentrations was performed by converting the galectin concentrations into Z-scores, adding these together pairwise, and running the appropriate test as described above. Values of *p* less than 0.05 were considered statistically significant. Data is available in Appendix A

## 3. Results

### 3.1. Patient Characteristics

Serum galectin concentrations were compared using patient characteristics: ethnicity, race, and smoking history. The *p*-values of these analyses are found in Table 1. As all the patients in this sample were female, no comparison was able to be made by sex. No statistically significant changes in the galectin concentrations were observed based on patient characteristics, as shown previously [32].

### 3.2. Stage

Figure 1 shows the differences in galectin concentrations between different stages and subtypes of breast cancer. No significant differences were discovered for galectins-3 and -9 concentrations in either stage or subtype of breast cancer. However, galectin-1 concentration was statistically significantly elevated in stage III breast cancer compared to stages I and IV. When grouped by stage and molecular subtype, galectin-1 concentration was statistically significantly elevated in stage III luminal A breast cancer compared to stage I.

### 3.3. Molecular Data

The level of hormone-receptor expression does not correlate with galectin serum concentrations, as seen in Figure 2. While the specific level of hormone expression in breast cancer does not typically guide breast cancer treatment, its relationship with serum galectin concentrations is reported here for the benefit of future basic science investigations.

Figure 3 shows the differences in galectin concentrations between any expression versus non-expression of either hormone receptor, estrogen receptor and progesterone receptor separately, or HER2 amplification. Galectin-9 showed statistically significant changes in serum concentrations for all three molecular markers, increasing in HER2 amplified samples while decreasing in HR positive samples. There were no statistically significant findings for galectins-1 and -3.

### 3.4. Oncotype

Figure 4 shows the differences in galectin concentrations between different molecular subtypes of breast cancer. No significant difference was discovered for galectins-1 and -3; however, galectin-9 concentrations showed a statistically significant increase for the HER2-enriched subtype.

### 3.5. Treatment Status

Figure 5 shows galectin concentrations based on patients’ status of having received systemic therapy vs. no therapy prior to serum collection. Some patients may have received neoadjuvant or adjuvant systemic therapy, depending on their time of diagnosis. Serum galectin-1 (Figure 5A) was statistically significantly increased in patients who had some form of systemic therapy compared to patients that had not received treatment. When galectin-1′s concentrations are grouped by molecular subtype (Figure 5C), Luminal A subtype again has a statistically significant increase of galectin-1 level in patients who received systemic therapy.

### 3.6. Combination Analysis

An analysis of the combination of galectin concentrations compared to stage, molecular subtype, and receptor expression was performed. The results of this analysis are seen in Table 2. The only significant result was the combined galectin-1 and galectin-3 Z-score based on the stage of breast cancer.

Appendix A provides a summary of the serum galectin values observed across various characteristics in the data and can be found in Appendix A.

## 4. Discussion

### 4.1. Findings

There were no changes in galectin concentrations by stage, as seen in Figure 1, except for galectin-1 in stage III compared to stages I and IV. While galectins-1, -3, and -9 have been established to be increased in the serum of breast cancer patients, there does not appear to be any appreciable pattern in these lectins by stage [32]. This negative finding suggests that alterations in the levels of galectins-1, -3, and -9 are not predictive of breast cancer stage. With higher stages of breast cancer conferring worse prognosis, the prognostic abilities of galectins-1, -3, and -9 may be limited in breast cancer.

There was no correlation between the specific serum galectin concentrations and hormone receptors’ specific level of expression in breast tissue (Figure 2). ER receptor binding has been shown to induce the expression of galectin-3 in prostate cancer [39], and expression of galectins-1, -3, and -9 have been shown to be correlated with binary levels of HR expression in breast cancer tissue [36]. Therefore, this negative finding of non-correlation with galectin serum concentrations is valuable, as it also can help inform future studies and biomarker searches.

Conversely, the galectins’ serum concentrations were compared to receptor expression or amplification. Figure 3 shows that galectin-9 had decreased serum concentrations in samples with positive hormone receptor markers and increased serum concentrations in HER2-amplified tissues. The relationship between ER status and galectin-9 has been examined in one study, which found that galectin-9 expression did not correlate with ER status. However, there was a trend showing that ER-positive breast cancers were more likely to be galectin-9-negative compared to ER-negative breast cancers [40]. This study found the same relationship in the patient serum and highlights the importance of understanding the implications of galectin-9 concentrations in a breast cancer patient. For example, levels of galectins-7 and -8 have shown relationships with the breast cancer receptor status and have evidence suggesting potential uses as independent prognosticators for impaired progression-free survival [41]. Interestingly, galectin-3 has not shown any relationship in this regard [42]. Unlike ER status, there have been no direct links between galectin-9 and HER2 specifically, but galectin-9 is an emerging biomarker for breast cancer invasiveness [43,44].

While the concentrations of these three galectins are known to be increased in breast cancer and the differences between galectin expression in breast cancer tissue has been discussed [36], the difference in serum galectin concentrations based on molecular subtype has not been previously reported. As seen in Figure 4, levels of galectin-9 were observed to be higher in HER2-enriched breast cancer patients than the other three molecular subtypes, while there were no differences in serum concentrations of galectins-1 and -3 between the subtypes. This finding is congruent with the previous finding, described in Section 3.3, regarding galectin-9’s decreased serum concentrations in samples with positive hormone-receptor markers but an increase in serum concentrations in HER2-amplified tissues. Since HER2-enriched breast cancer is defined based on amplified HER2 and negative for HR, this seems to be an intuitive outcome. This intriguing finding warrants deeper investigation through targeted studies focused on exploring galectin-9’s connection with HER2-enriched breast cancer compared to other molecular subtypes.

Galectin-1 concentrations were found to be statistically significantly increased when cancerous tissues are exposed to cancer therapies other than resection or radiation. Therapies that samples could have received included chemotherapy, hormonal blockade, and immunotherapy. This represents the first demonstration of increasing galectin-1 expression in the setting of breast cancer treatments. Since the serum galectin levels are a picture of the whole body’s response, not just the cancer, this increase in galectin-1 could represent a rise based on an inflammatory response to the therapies used.

Finally, in the analysis of the combination of the galectin levels, no significant differences were found except for the combination of galectin-1 and galectin-3 based on the stage of breast cancer. However, the *p*-value for the ANOVA is greater than that of galectin-1 compared to the stage of breast cancer alone (0.0478 vs. 0.0253). The effect most likely comes from the galectin-1 values, and the addition of galectin-3 results in no significant signal or insight.

### 4.2. Limitations

This study has limitations based on the availability of samples for testing. The selection process for samples consisted of a random sampling of the PHCI’s biorepository of breast cancer patients. This selection process can make analysis difficult, as one is not guaranteed uniform comparison groups. However, it does provide an overview of the characteristics of a representative patient panel.

While it was interesting that galectin-1 increased in response to the exposure to any chemotherapy, hormonal therapy, or immunotherapy, it was not possible to retrieve information regarding the exposure of the tumor to specific therapeutic agents or treatment plans. Further delineation of the response of serum galectins based on the specific chemotherapy, hormonal agents, or immunotherapy was not able to be determined. This area needs to be further investigated, as the response of the galectins and their change in levels to specific therapies could provide clues as to the tumor response to therapy. Galectin levels may also change depending upon treatment response and may be of prognostic value.

Low galectin-9 has been shown to have significantly decreased overall survival (OS) in triple negative breast cancer tumor cells with PDL-1 negativity [45]. Therefore, the correlation between galectin concentrations and OS is of interest in oncology research. However, survival data was not available for our patient population; therefore, we were not able to assess galectin’s potential relationship to OS or progression-free survival in breast cancer.

## 5. Conclusions

These findings encompass a wide range of insights and will provide direction for future galectin research. Galectin-9′s increase in HER2-enriched breast cancers has the potential to be further delineated to provide a mechanistic understanding of the relationships, potential therapeutic targets, and use as indicators of disease. These findings are especially impactful in the context of the current interest in galectin-9 and its role in the tumor microenvironment regulating the immune response to cancer via TIM-3 [46,47]. This study’s findings also provide evidence that future galectin investigations need to be cognizant of the relationship between the treatment status of the patient and the level of circulating galectins. It is important to understand confounding variables such as patient treatment status that have the possibility of changing the inflammatory landscape, as these proteins often have intimate functions in both the immune and inflammation processes.

Many studies have focused on measuring galectin expression levels in cancer tissue. While measuring a protein marker at the tissue level is important to understand its influence and impact on the development and progression of cancer, a complete understanding of these proteins’ levels in the human serum is necessary to provide a holistic picture. Effective and available biomarkers will rely on the ease of access from the patient and the ease of measurement. Serum galectin proteins lend themselves to these ends by being readily accessible and measurable using standard lab equipment.

Changes in galectin-9 concentrations based on subtype provide an interesting area for future research. The mechanism of the increase in galectin-9 level in HER2-enriched cancers and the relationship between serum levels and hormone and HER2 receptor expression have the potential to provide more insight into tumor biology.

Moving forward, it will be important to understand the changes in galectin levels before and after resection. While serum galectins have the advantage of being easily collected and measured, while providing possible insights into the tumor’s cancer biology, the fact that galectins are a part of a wide variety of biological processes introduces confounding factors. A way to better understand the relationship between serum galectin levels and cancer biology would be to measure galectin levels pre- and post-resection of the tumor, as it would better provide a more nuanced understanding of the mass’s contribution to the serum galectin load.

## Figures and Tables

**Figure 1 cancers-15-03809-f001:**
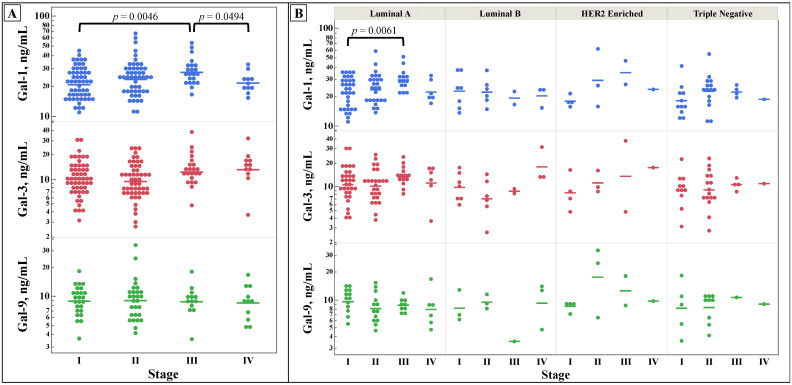
Comparison of serum galectins-1, -3, and -9 concentrations, as determined by ELISA, between stages of breast cancer (**A**) and by stage and molecular subtype (**B**). Multiple comparisons were performed with one-way ANOVA, with subsequent analysis using all pairs using Student’s *t*-test if the ANOVA’s probability > F was less than 0.05.

**Figure 2 cancers-15-03809-f002:**
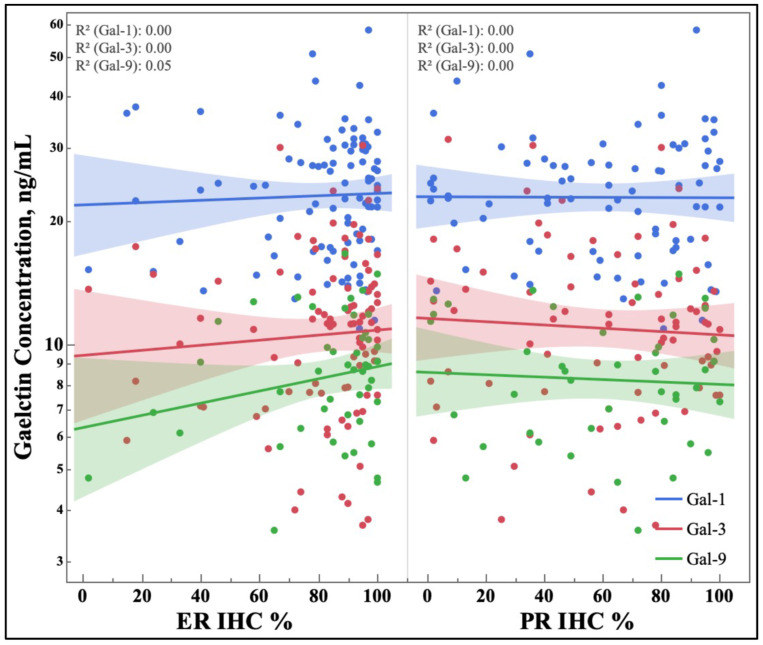
Graph of serum galectins-1, -3, and -9 concentrations, determined by ELISA, based on the level of estrogen and progesterone receptor expression in breast cancer via immunohistochemical stain. Blue, Gal-1; Red, Gal-3; Green, Gal-9; ER, Estrogen Receptor; PR, Progesterone Receptor; IHC, Immunohistochemistry.

**Figure 3 cancers-15-03809-f003:**
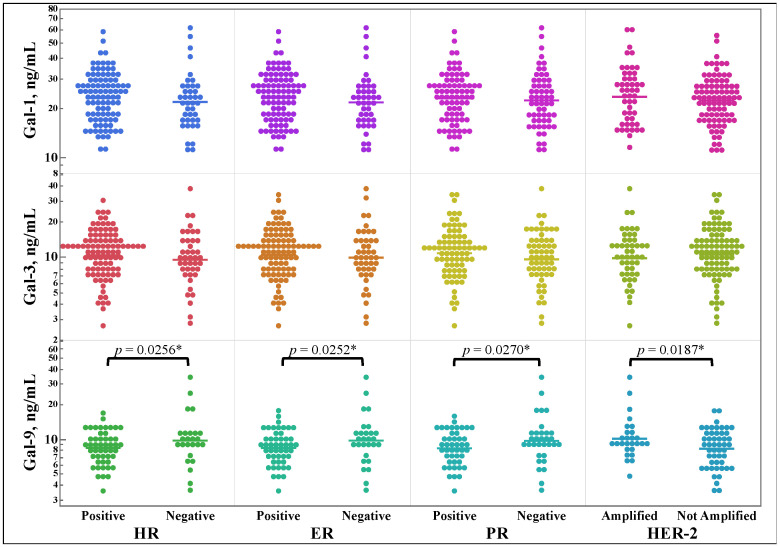
Comparison of serum galectin-1, -3, and -9 concentration in breast cancer patient serum, as determined by ELISA, by binary expression of overall hormone receptor, estrogen and progesterone receptor, and HER2 amplification determined using an immunohistochemical stain. HR, Hormone Receptor; ER, Estrogen Receptor; PR, Progesterone Receptor. * One-tail *t*-test.

**Figure 4 cancers-15-03809-f004:**
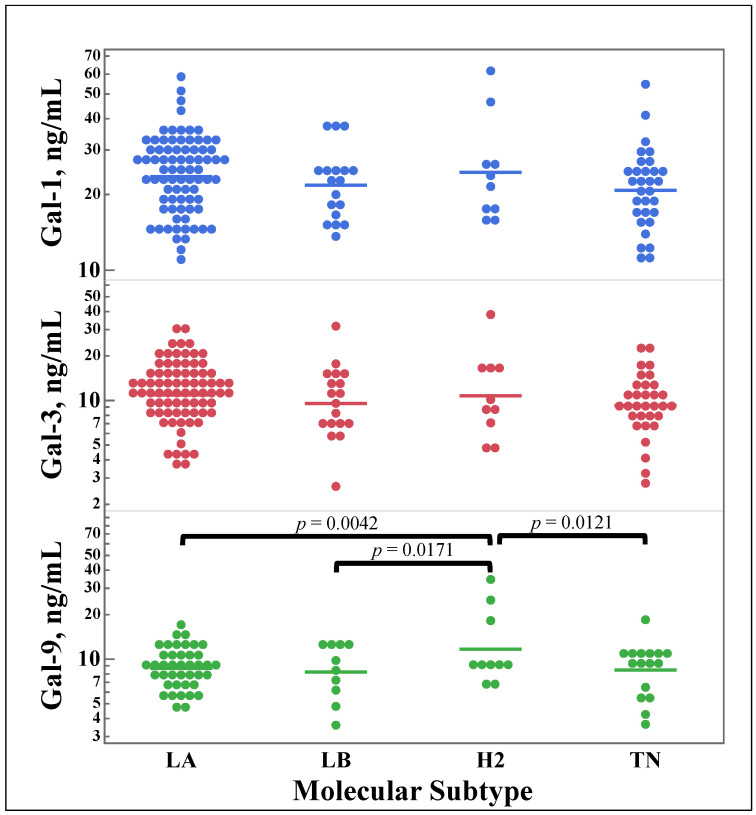
Comparison of serum concentrations of galectins-1, -3, and -9 between molecular subtypes of breast cancer. Galectin concentrations were measured from serum samples of breast cancer patients using ELISA assays. Patients were classified using the molecular subtype of breast cancer. Multi-pair analyses of galectins based on subtype were carried out using Student’s *t*-test. LA, Luminal A; LB, Luminal B; H2, HER2-enriched; TN, Triple Negative.

**Figure 5 cancers-15-03809-f005:**
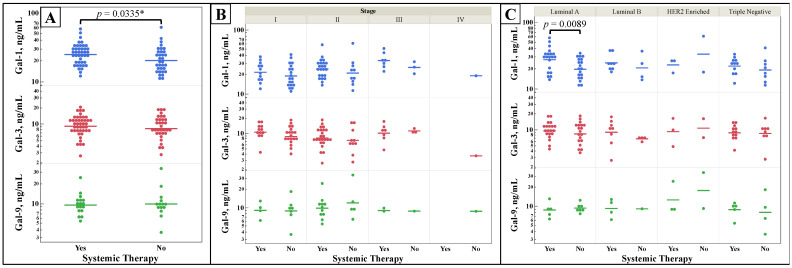
Comparison of galectins-1, -3, and -9 concentrations based on patient exposure to immunological, chemotherapy, or hormone therapy, “Systemic Therapy”. Galectin concentrations were measured from serum samples of breast cancer patients by ELISA assays. Patients were classified using the status of receiving any form of immunological, chemotherapy, or hormone therapy. The serum concentrations of galectins of patients who were exposed vs. not exposed to systemic therapy were compared alone (**A**), by stage (**B**), and by molecular subtype (**C**). Two-group comparisons were made by *t*-test. Multiple comparisons were performed with one-way ANOVA, with subsequent analysis of all pairs using Student’s *t*-test if the ANOVA’s probability > F was less than 0.05. * One-tail *t*-test.

**Table 1 cancers-15-03809-t001:** *p*-values of Serum Galectin Concentrations by Patient Characteristics.

Attribute	Gal-1	Gal-3	Gal-9
Ethnicity	0.6827	0.5913	0.3999
Race	0.7846	0.3277	0.8295
Smoking History	0.5084	0.9730	0.5485

**Table 2 cancers-15-03809-t002:** *p*-values of Combined Galectin Z-Scores Based on the Breast Cancer Characteristic.

Characteristic	Gal-1 + Gal-3	Gal-1 + Gal-9	Gal-3 + Gal-9	Gal-1 + Gal-3 + Gal-9
Stage	0.0478	0.7774	0.7774	0.8034
Molecular Subtype	0.2019	0.0965	0.0965	0.2471
HR Expression	0.6021	0.2143	0.2143	0.4991
ER Expression	0.6707	0.2779	0.2779	0.5200
PR Expression	0.3904	0.2145	0.2145	0.6658
HER2 Amplification	0.8720	0.0973	0.0973	0.3268

Gal, Galectin; HR, Hormone Receptor; ER, Estrogen Receptor; PR, Progesterone Receptor.

## Data Availability

The data presented in this study are available in Appendix A.

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
