# Peer review of "Pattern Analysis of Serum Galectins-1, -3, and -9 in Breast Cancer"

_cancers, 2023, doi:10.3390/cancers15153809_

Round 1
Reviewer 1 Report
In this study, the authors investigated pattern analysis of Galectin 1, 3 and 9 Serum Levels in Breast Cancer, by enzyme-linked immunosorbent assay (ELISA). The study included a series of 137 breast cancer patient serum samples well characterized for clinical-pathologic characteristics as well as patient smoking history, tissue exposure to oncological treatment, and subtype classification (Luminal A, Luminal B, HER-2 Enriched, and Triple Negative). The authors also performed the comparison among Galectin 1, 3 and 9 Serum Levels and the characteristics of the patient and tumor such as stage, molecular subtype, and receptor expression. From this analysis, Galectin-9 was found to be statistically significantly increased in HER2 enriched tumors and reduced in patients with hormone receptor positive tumors; Galectin-1 was found to be statistically significantly increased in the serum of patients who had undergone hormonal, immunotherapy, or chemotherapy.
Authors conclude that the relationship between galectin and tumor biology could be used to inform future research on galectin biomarkers and as therapeutic target using galectin inhibitors.
Main points:
- The authors should specify also in the text of the manuscript the number of samples for which they have galectin 1, 3 and 9 levels. Moreover, the authors should specify the number of patients in the comparison performed (between galectin levels with clinical-pathologic characteristics and with treatment status) and better describe if they include all patients in all comparisons.
- In Materials and Methods section, the authors should better describe the information about therapy of the patients before and after serum collection.
- The authors should check the correspondence of number of patients inserted in the text of the manuscript and in the Table 1 (supplementary materials) for each characteristic. For example, from line 108 to 113 of Materials and Methods, the number of patients for the classification in Luminal A, Triple Negative and of stage II do not correspond to the number in Table 1 for the same characteristics.
- In the Results section (3.1 Stage), the authors should clearly describe also the statistically significant results regarding the comparison between galectin levels and stages broken down by molecular subtypes.
Minor points:
- In Materials and Methods section, the authors described the breast cancer case series including 137 serum samples, but in the excel file called “Data sheet” there are information for 138 samples.
- In the text of the manuscript the authors should check the abbreviations used (for example, the abbreviations used for estrogen and progesterone receptor).
- In the abstract section, at the lines 25 and 26, the authors could delete the sentence “and increased in patients with HER2 amplified tissues”, because it is inserted twice (also at the beginning of the sentence, line 24 and 25).
Overall, the manuscript needs some English language editing.
Author Response
Response to Reviewer One
We are writing in response to your valuable observations and feedback on our study. We genuinely appreciate your time and thoroughness in reviewing our manuscript. Please find below our responses to your points reproduced in bold for your convenience.
- The authors should specify also in the text of the manuscript the number of samples for which they have galectin 1, 3 and 9 levels. Moreover, the authors should specify the number of patients in the comparison performed (between galectin levels with clinical-pathologic characteristics and with treatment status) and better describe if they include all patients in all comparisons.
We agree this would provide more clarity. We have added to sub-section 2.1 of the materials and methods the total number of galectin-1, -3, and -9 serum samples obtained. Additionally, we have described the number of patients with each clinical-pathologic characteristic with measured galectin levels in the supplementary Table S1 and provided language (page 3, lines 121-122) in the manuscript directing the reader to this table. Thank you allowing us to provide clarity.
- In Materials and Methods section, the authors should better describe the information about therapy of the patients before and after serum collection.
We agree that the materials and methods could be expanded upon regarding the treatment of the patients before and after sample collection. Thus, we have better described systemic therapy and the treatments, or lack thereof, that patients received using this definition. Additionally, we mention how treatments after sample collection were, unfortunately, not available (page 3, lines 105-106). We feel that this now better correlates with the limitations section of our paper (page 9, lines 389-394), and we thank you for your assistance in improving our manuscript.
- The authors should check the correspondence of number of patients inserted in the text of the manuscript and in the Table 1 (supplementary materials) for each characteristic. For example, from line 108 to 113 of Materials and Methods, the number of patients for the classification in Luminal A, Triple Negative and of stage II do not correspond to the number in Table 1 for the same characteristics.
This is an astute observation, and we recognize that the manuscript was confusing in its previous construction. These apparent discrepancies appeared to occur, because not every patient with Luminal A, TNBC, and stage II breast cancer had their serum samples of galectins 1, 3, and 9 quantified. Thus, for example, there were 32 patients with TNBC in our study, but galectin-1 levels were obtained in only 31 of those, galectin-3 obtained in all 32, and galectin-9 in only 16. Due to this confusion, the materials and methods were expanded upon to alleviate confusion (page 3, lines 106-107, 128-129, and 133-135) and the changes made in the previous comment should assist in this as well. Thank you for the thoroughness of your review as it has increased the coherence of the manuscript.
- In the Results section (3.1 Stage), the authors should clearly describe also the statistically significant results regarding the comparison between galectin levels and stages broken down by molecular subtypes.
The results section under section 3.2 (formerly 3.1) has been expanded upon with more digestible discussion of the statistically significant findings for galectin-1 as it relates to stage and molecular subtype, specifically Luminal A breast cancer (page 4, lines 154-158). Additionally, the negative findings regarding galectins-3 and -9 are more clearly stated (page 4, lines 153-154). Thank you for your review of our manuscript.
- In Materials and Methods section, the authors described the breast cancer case series including 137 serum samples, but in the excel file called “Data sheet” there are information for 138 samples.
There were 139 total samples as seen in this data sheet. Thus the ‘137’ was a typo that has been corrected. Due to that mistake, subsequent corrections were made in the supplemental materials and throughout the manuscript. The data now accurately correlates from manuscript to supplemental materials and the data sheet. Additionally, since not all values for each patient were documented, we have added “Undocumented” as an entry into Table S1 to show that there is no discrepancy in the number of samples but that some samples did not arrive with the information. We apologize for these clerical mistakes, and we appreciate your review of our paper.
- In the text of the manuscript the authors should check the abbreviations used (for example, the abbreviations used for estrogen and progesterone receptor).
The abbreviations estrogen receptor (ER), progesterone receptor (PR), and hormone receptor (HR) were examined throughout the manuscript to ensure there were no inaccuracies or misspellings. Thank you for ensuring accurate abbreviations in our manuscript.
- In the abstract section, at the lines 25 and 26, the authors could delete the sentence “and increased in patients with HER2 amplified tissues”, because it is inserted twice (also at the beginning of the sentence, line 24 and 25).
To reduce redundancy, the sentence was deleted as suggested. We appreciate all the feedback you have provided on our manuscript.
We are sincerely grateful for your thorough review and thoughtful feedback on our manuscript. Your observations and suggestions have been instrumental in improving the clarity and accuracy of our study. Thank you once again for your time, dedication, and valuable input.
Please do not hesitate to reach out if you have any further questions or require additional information.
Warmest regards,
Avery Funkhouser
Reviewer 2 Report
The manuscript investigates galectin protein levels in the serum of breast cancer patients and their implications for diagnosis and treatment. Serum samples from breast cancer patients were analyzed, focusing on galectin-1, -3, and -9. The study finds that galectin levels do not significantly vary based on breast cancer stage, except for galectin-1 in stage III. While galectin-1, -3, and -9 are increased in the serum of breast cancer patients, there is no consistent pattern across stages. The correlation between galectin levels and hormone receptor expression in breast tissue is non-existent, except for galectin-9, which shows decreased levels in samples with positive hormone receptors and increased levels in HER2-amplified tissues. Galectin-9 is also higher in HER2-enriched breast cancer compared to other subtypes. Notably, galectin-1 levels increase in response to treatments other than resection or radiation. The study acknowledges limitations in sample selection and the lack of specific therapeutic agent information. The authors conclude that further investigation is needed for galectin-9 in HER2-enriched breast cancer to understand its mechanisms and identify therapeutic targets. They emphasize considering treatment status when studying circulating galectin levels and highlight the potential of serum galectins as accessible biomarkers. The manuscript provides insights into galectin levels in breast cancer patients' serum and suggests future research directions and clinical applications.
To further enhance the manuscript, the following missing content could be included:
1) Patient outcomes: Any available information on patient follow-up, including treatment response, overall survival, and progression-free survival. Analyze the correlation between galectin levels and these outcomes to assess their prognostic value.
2) Treatment-specific analysis: Separate the patient cohort into two groups based on whether they received treatment before sample collection. Analyze galectin levels separately for treated and untreated patients to evaluate the correlation between galectin expression and certain characteristics.
3) Additional patient characteristics: Investigate the correlation between galectin levels and other patient characteristics, such as sex, ethnicity, race, and smoking history. Explore potential associations or differences based on these factors.
4) Comparative analysis: Include figures illustrating the comparison of galectin-1 and -3 levels among different breast cancer subtypes. Also, show the comparison of galectin-1 and -3 levels based on binary expression of estrogen and progesterone receptors and HER2 amplification.
5) Hormone receptor analysis: Instead of analyzing estrogen receptor (ER) and progesterone receptor (PR) expression separately, consider combining them into hormone receptor (HR) status. Examine the correlation between HR status and galectin levels to provide a comprehensive understanding of hormone receptor-related associations.
6) Combination analysis: Explore the correlation between combined galectin-1, -3, and -9 levels and their relationship with breast cancer subtypes, hormone receptor status, and HER2 amplification. Assess whether combined galectin measurements provide additional insights compared to individual galectin analysis.
By incorporating these missing content areas, the manuscript will offer a more comprehensive analysis and provide valuable insights into the relationship between galectin levels and various clinical factors in breast cancer patients.
Author Response
Response to Reviewer Two
We are writing in response to your valuable observations and feedback on our study. We genuinely appreciate your time and thoroughness in reviewing our manuscript. Please find below our responses to your points reproduced in bold for your convenience.
In response to your observations, we would like to address the following points:
1) Patient outcomes: Any available information on patient follow-up, including treatment response, overall survival, and progression-free survival. Analyze the correlation between galectin levels and these outcomes to assess their prognostic value.
We recognize that galectin levels may have prognostic value within breast cancer, however we do not have access to overall survival (OS), progression free survival (PFS), or treatment response after serum sample collection. Due to this we are unable to perform analysis to assess this, but we have expanded upon this in the limitations of this manuscript (page 9, lines 297-300; page 3, lines 105-106). We appreciate your evaluation and review of our manuscript.
2) Treatment-specific analysis: Separate the patient cohort into two groups based on whether they received treatment before sample collection. Analyze galectin levels separately for treated and untreated patients to evaluate the correlation between galectin expression and certain characteristics.
Tissue samples may have been exposed to chemotherapy, hormonal therapy, or immunotherapy. These three treatment possibilities are defined as systemic therapy, and this is now more clearly defined in the materials and methods (page 3, lines 102-106). Our study found that galectin-1 concentration is statistically significantly increased when patients received systemic therapy before serum sample collection as demonstrated in Figure 5. These clarifications in the Methods section have made the Results section more cohesive and easier to follow along with some minor adjustments made in the Results section as well. Thank you for your review, we appreciate how it has improved the quality of our paper.
3) Additional patient characteristics: Investigate the correlation between galectin levels and other patient characteristics, such as sex, ethnicity, race, and smoking history. Explore potential associations or differences based on these factors.
Previously our data was analyzed to identify any significant changes to galectin levels based upon demographic characteristics. There are no statistically significant findings, but these negative findings have been added to our manuscript (pages 3 and 4, section 3.1) to provide a more comprehensive description of our results. We appreciate your evaluation of our manuscript and its subsequent improvement.
4) Comparative analysis: Include figures illustrating the comparison of galectin-1 and -3 levels among different breast cancer subtypes. Also, show the comparison of galectin-1 and -3 levels based on binary expression of estrogen and progesterone receptors and HER2 amplification.
We have reconstructed the previous Figures 3 and 4 to include galectins-1 and 3. Figure 3 now includes galectins-1 and -3 levels stratified by HR, ER, PR, and HER2 binary amplification, and Figure 4 now includes galectins-1 and -3 levels stratified by molecular subtype. This data was not statistically significant, but this provides a more complete picture regarding galectins-1, -3, and -9 levels delineated by hormone receptor, HER2 status, and molecular subtype. Thank you for your evaluation of our manuscript.
5) Hormone receptor analysis: Instead of analyzing estrogen receptor (ER) and progesterone receptor (PR) expression separately, consider combining them into hormone receptor (HR) status. Examine the correlation between HR status and galectin levels to provide a comprehensive understanding of hormone receptor-related associations.
We have added hormone receptor status into Figure 3 to provide a better understanding of the associations. Combined hormone receptor status defined as binary positivity for ER, PR, or both. An explanation of this was added to the Methods (page 3, line 107-109). Thank you for this suggestion.
6) Combination analysis: Explore the correlation between combined galectin-1, -3, and -9 levels and their relationship with breast cancer subtypes, hormone receptor status, and HER2 amplification. Assess whether combined galectin measurements provide additional insights compared to individual galectin analysis.
Thank you for this feedback, the combination of the galectin values is an interesting idea and one that we were excited to perform. To standardize the galectin values, we converted their respective concentrations into Z-scores and added these scores together pairwise to combine the values (page 3, lines 137-140). These values were then analyzed by stage, subtype, and hormone levels as suggested. We share the results in a new section and table (Section 3.5, Table 2). It does not appear that the combination of galectin values provides any new insights (discussed on page 9, lines 276-280), however, the additional analysis provides a more robust analysis and discussion, thank you for your suggestion.
We are sincerely grateful for your thorough review and thoughtful feedback on our manuscript. Your observations and suggestions have been instrumental in improving the clarity and accuracy of our study. Thank you once again for your time, dedication, and valuable input.
Please do not hesitate to reach out if you have any further questions or require additional information.
Warmest regards,
Avery Funkhouser